# The Use of Star Anise-Cinnamon Essential Oil as an Alternative Antibiotic in Prevention of *Salmonella* Infections in Yellow Chickens

**DOI:** 10.3390/antibiotics11111579

**Published:** 2022-11-09

**Authors:** Changcheng Li, Ziheng Xu, Wenyan Chen, Chenyu Zhou, Can Wang, Min Wang, Jingzhen Liang, Ping Wei

**Affiliations:** 1Institute for Poultry Science and Health, Guangxi University, Nanning 530004, China; 2School of Public Health and Management, Guang University of Chinese Medical, Nanning 530200, China

**Keywords:** star anise-cinnamon essential oil, trans-anethole, trans-cinnamaldehyde, antibiotic alternatives, *Salmonella pullorum*, *Salmonella give*, *Salmonella kentucky*, prevention

## Abstract

*Salmonella* is capable of harming human and animal health, and its multidrug resistance (MDR) has always been a public health problem. In addition, antibiotic-free or antibiotic-reduced policies have been implemented in poultry production. Therefore, the search for antibiotic alternatives is more urgent than ever before. The aim of this study was to assess the antibacterial activity of star anise-cinnamon essential oil (SCEO) in vitro and its prophylactic effect against the infections of *Salmonella pullorum*, *Salmonella give*, and *Salmonella kentucky* in vivo. The results demonstrated that SCEO is effective against *Salmonella pullorum*, *Salmonella give*, and *Salmonella kentucky* in vitro. Supplementation with SCEO could significantly decrease the infections of *Salmonella pullorum* and *Salmonella give*, whereas it could slightly but not significantly decrease the infection of *Salmonella kentucky*, while also significantly alleviating the body weight (BW) loss caused by the infections of *Salmonella pullorum*, *Salmonella give*, and *Salmonella kentucky* in Yellow chickens. The SCEO had the best prophylactic effect against the infection of *Salmonella give* in Yellow chickens, followed by the infection of *Salmonella pullorum* and the infection of *Salmonella kentucky*. The SCEO, used as an antibiotic alternative, could be an effective prevention strategy against the infections of *Salmonella pullorum*, *Salmonella give*, and *Salmonella kentucky* in Yellow chickens.

## 1. Introduction

*Salmonella* is an important zoonotic pathogen capable of severely endangering public health; infections of different serotypes of *Salmonella* in humans and animals lead to a variety of diseases [1]. For a long time, *Salmonella* infection in chickens has been very harmful to the poultry industry. Pullorum disease (PD), caused by *Salmonella pullorum*, is a major *Salmonella* disease that seriously harms the health of poultry flocks. Paratyphoid infections (PIs) are caused by paratyphoid Salmonellae, such as *Salmonella give* and *Salmonella kentucky*. The harm of PIs to poultry flocks cannot be underestimated; however, PIs have mainly received extensive attention in terms of foodborne outbreaks of human illness. Yellow chickens have an important role in the China poultry industry with annual sales of approximately 4.3 billion in 2021, representing approximately 43% of the total annual sales of broilers [2,3]. However, most chicken farms in China are still threatened by *Salmonella*. Birds infected with *Salmonella pullorum* or paratyphoid Salmonellae without effective elimination can lead to reduced production performance and even death. Furthermore, infected birds transmit *Salmonella* to other healthy birds through horizontal and vertical transmission, creating a vicious cycle [4]. This has brought serious economic losses to the poultry industry [5]. Due to the implementation of the National Poultry Improvement Program (NPIP), *Salmonella pullorum* has been eradicated from commercial poultry in developed countries, but it is still prevalent in developing countries [4]. Although paratyphoid Salmonellae, such as *Salmonella give* and *Salmonella kentucky*, are less harmful to the poultry industry than *Salmonella pullorum* and are not a part of the NPIP, they also cause public health events through poultry products occasionally [6,7]. In our previous epidemiological investigation from 2019 to 2022, *Salmonella pullorum* (87.37%, 512/586) was the most dominant *Salmonella* serotype isolated in Yellow chickens, as identified by our research group [8,9,10]. Other paratyphoid Salmonellae, such as *Salmonella kentucky* (4.78%, 28/586) and *Salmonella give* (4.44%, 26/586), were also isolated from Yellow chicken on farms in recent years [8,9,10]. Both *Salmonella kentucky* and *Salmonella give* are *Salmonella enterica* serovars that infect humans; thus, they may become common food-borne bacteria.

At present, *Salmonella* infections in the poultry industry are mainly prevented and controlled by antibiotics. However, the irrational use of antibiotics has led to a gradual increase in the antibiotic resistance rate of *Salmonella* and an increasingly serious phenomenon of multidrug resistance (MDR) [11]. This will not only prolong the illness duration and increase the cost, but also enhance the risk of death. The most worrying fact is that the development of new antibiotics to fight diseases lags far behind the emergence of antibiotic resistance [12]. This will make avian salmonellosis increasingly difficult to effectively prevent and control. Antibiotic-free or antibiotic-reduced policies have been implemented in poultry production. Many countries around the world have promulgated and implemented policies to ban the use of feed antibiotics. China also banned the use of feed antibiotics in 2020 [13]. Therefore, the search for and development of feed antibiotic alternative represent a current research hotspot. 

Plant essential oils are a new, safe, efficient, and green natural compounds. They are mainly extracted from the flowers, leaves, stems, roots, or fruits of plants by distilling with water vapor. They are volatile aromatic oily liquids, which contain a variety of components, such as alcohols, olefins, esters, ketones, and aldehydes [14]. Plant essential oils have the advantages of antimicrobial, anti-inflammatory, antioxidant, and growth-promoting activities. Therefore, plant essential oils represent an efficacious nutritional alternative strategy to prevent/control intestinal diseases and to reduce the negative impacts of diseases. Star anise-cinnamon essential oil (SCEO) is a plant essential oil extracted from star anise and cinnamon. SCEO is mainly composed of two monomers: trans-anethole and trans-cinnamaldehyde. The antimicrobial activities of trans-anethole and trans-cinnamaldehyde have already been proven [15,16]. Dietary supplementation with star anise essential oil can improve the growth performance, carcass characteristics, and meat quality of broiler chickens [17]. Cinnamon essential oil supplementation could inhibit the growth of harmful intestinal pathogens and improve the average daily gain of broilers [18]. Furthermore, combining more than two plant essential oils might, in general, be superior to the use of a single plant essential oil.

However, no study to date has reported the prevention of *Salmonella* infection in Yellow chicken using SCEO. Therefore, the objectives of the present study were to evaluate the antibacterial activity of SCEO in vitro and its beneficial effects against the infections of *Salmonella pullorum*, *Salmonella give*, and *Salmonella kentucky* in commercial Yellow chicken breeders.

## 2. Results

### 2.1. Minimum Inhibitory Concentration (MIC) of SCEO

Table 1 shows the MIC of SCEO against *Salmonella pullorum*, *Salmonella give*, and *Salmonella kentucky*. The MIC of the SCEO for *Salmonella pullorum* was 1:8. For *Salmonella give* and *Salmonella kentucky*, those dilutions were lower, reaching 1:2.

### 2.2. Clinical Signs and Gross Lesions

Birds in groups A and B exhibited weakness, poor growth (Figure 1a), white diarrhea (Figure 1b), and wet anuses (Figure 1c), while the birds in group C had none of these symptoms. The highest number of birds with these symptoms was recorded in group B. At necropsy, hearts with no lesions (Figure 1d), heart with granulomas (Figure 1e), soft hearts with pericardial effusion (Figure 1f), livers with no lesions (Figure 1g), and livers with pinpoint white foci of necrosis (Figure 1h) could be observed in group B. Only one death case was recorded during the experiment, which occurred in subgroup B1.

### 2.3. Body Weight (BW)

Results of the BW test are demonstrated in Figure 2. Appendix A shows the raw data of BW test. In the *Salmonella pullorum*-challenged groups, the average BW of birds in subgroup A1 was significantly higher than in subgroup B1 (*p* < 0.05). In the *Salmonella give*-challenged groups, the average BW of birds in subgroup A2 was significantly higher than in subgroup B2 (*p* < 0.05). In the *Salmonella kentucky*-challenged groups, the average BW of birds in subgroup A3 was significantly higher than in subgroup B3 (*p* < 0.05). Furthermore, the average BW of group C was significantly higher (*p* < 0.05) than in other groups, but no significant difference (*p* > 0.05) was observed between group C and subgroup A2, although that of group C was slightly higher than that of subgroup A2.

### 2.4. The Recovery Rate of the Challenged Salmonella in Birds

If *Salmonella* was detected in any sample of bird, it was determined that the bird was successfully infected with *Salmonella*. The infection rate of *Salmonella* in each group is shown in Figure 3 and Appendix A. In the *Salmonella pullorum*-challenged groups, the infection rate of *Salmonella pullorum* in subgroup A1 was significantly lower (*p* < 0.05) than in subgroup B1. The same trend was observed in the *Salmonella give*-challenged groups (*p* < 0.05). In the *Salmonella kentucky*-challenged groups, the infection rate of *Salmonella kentucky* in subgroup A3 was slightly and not significantly lower (*p* > 0.05) than in subgroup B3. No *Salmonella* was isolated from the group C birds.

Figure 4a shows the isolations of *Salmonella* from different samples of birds in challenged/untreated groups. In the *Salmonella pullorum*-challenged/untreated subgroup B1, the isolations of *Salmonella pullorum* in the samples of the mixed organs and cecum were slightly but not significantly higher (*p* > 0.05) than in cloaca swabs. The same trend was observed in the *Salmonella give*- and *Salmonella kentucky*-challenged/untreated subgroups B2 and B3. Figure 4b–d show the isolations of *Salmonella* from different samples of birds in challenged groups. Appendix A show the detailed situations of *Salmonella* isolation from different organs. In the *Salmonella pullorum*-challenged groups, the isolations of *Salmonella pullorum* from samples of mixed organs, cecum, and cloaca swabs in subgroup A1 were slightly but not significantly lower (*p* > 0.05) than in subgroup B1. The same trend could also be observed in the *Salmonella give*-challenged groups. In the *Salmonella kentucky*-challenged groups, the isolation of *Salmonella kentucky* from samples of mixed organs in subgroup A3 was slightly but not significantly lower (*p* > 0.05) than in subgroup B3, whereas the isolations of *Salmonella kentucky* from samples of cecum and cloaca swabs in subgroup A were slightly and not significantly higher (*p* > 0.05) than in subgroup B3. 

## 3. Discussion

Avian salmonellosis is an infectious disease that seriously endangers the development of poultry industry in China [4]. Under the background of antibiotic-free breeding, plant essential oils are an antibiotic alternative with good antibacterial activity. Our experimental results showed that infections of *Salmonella pullorum*, *Salmonella give*, and *Salmonella kentucky* could cause severe gross lesions, significant weight loss and even death in Yellow chickens. Research has reported that *Salmonella pullorum* usually causes acute systemic disease in chicks, leading to the death of chicks. The mortality caused by *Salmonella pullorum* increases 0% to 100%, and the peak of death usually occurs at 2–3 weeks old. Chicks that survive early infection become inapparent *Salmonella pullorum* carriers with high probability, and they show obvious clinical signs and decreased production performance. Older poultry infected with *Salmonella pullorum* usually do not show any symptoms, but PD occasionally causes the death of older poultry. The *Salmonella pullorum* carrier rate of eggs produced by older poultry infected with *Salmonella pullorum* is as high as 33%, and the embryo mortality or the rapid death among newly hatched birds can be seen after hatching [19]. In addition to *Salmonella pullorum*, paratyphoid Salmonellae also have considerable pathogenicity to chicks. Paratyphoid Salmonellae rarely cause acute systemic disease in chicks, but they can cause mass death in the highly susceptible chicks under stress conditions. The morbidity and mortality of chicks were higher during the first 2 weeks of life, with significant growth retardation. Older poultry infected with paratyphoid Salmonellae usually do not show any symptoms, but they can contaminate other healthy birds through horizontal and vertical transmission [19]. Therefore, *Salmonella give* and *Salmonella kentucky* isolated from Yellow chickens are not only a possible source of human food-borne bacterial infection, but are also harmful to farm poultry.

In vitro, the results of the present study indicated that the SCEO is effective against three *Salmonella* serotypes, exerting a stronger antibacterial activity on *Salmonella pullorum* than on *Salmonella give* and *Salmonella kentucky*. This seems to be extremely promising in terms of the control of salmonellosis in poultry breeding.

In vivo, the results of this study showed that SCEO could significantly decrease the infections of *Salmonella pullorum* and *Salmonella give*, while slightly but not significantly decreasing the infection of *Salmonella kentucky*. The effects of SCEO against the infections of *Salmonella pullorum* and *Salmonella give* were better than against the infection of *Salmonella kentucky*. On the one hand, the main components of SCEO are trans-anethole and trans-cinnamaldehyde, both of which have satisfactory antibacterial activity [15,20]. The antibacterial mechanism of trans-cinnamaldehyde mainly includes the following: (1) trans-cinnamaldehyde can disturb bacterial cell membrane structures, increase membrane permeability, and finally lead to the leakage of intracellular constituents [21]; (2) trans-cinnamaldehyde can inhibit membrane-bound ATPases of bacteria, disturb the proton motive force, block ATP synthesis, and finally lead to a reduction in bacterial energy sources [22]; (3) bacterial cell division is regulated by prokaryotic homologue filamentous temperature-sensitive protein Z (FtsZ), and trans-cinnamaldehyde can interfere with GTP to polymerize FtsZ into Z ring, which finally leads to inhibition of bacterial cell division [23]; (4) trans-cinnamaldehyde can reduce the expression of *luxR* and *bcsA* genes involved in the quorum sensing (QS) system and inhibit the synthesis of autoinducers (AIs) of the QS system, finally inhibiting the formation of biofilms by inhibiting the QS system [22]. The antibacterial mechanism of trans-anethole may be that trans-anethole disturbs the cell structure of bacteria, causing the leakage of intracellular constituents and the disorder of its anabolism, which finally leads to bacterial death [24]. On the other hand, trans-anethole also has a synergistic antibacterial effect. Research has shown that trans-anethole not only has an antibacterial effect, but also has synergistic effects on the antimicrobial activity of other antimicrobial agents against various microorganisms [24]. The synergistic antibacterial effect of trans-anethole greatly enhanced the antibacterial activity of trans-cinnamaldehyde, which is also the reason why this drug combination hinders the development of drug resistance in pathogens.

In order to effectively prevent and control *Salmonella* infection in birds, it is necessary to understand the pathogenic mechanism of *Salmonella* and the prevalence of *Salmonella* in various organs of birds. Cloaca swabs, cecum, and mixed organ samples were collected from the birds for the isolations challenged by *Salmonella pullorum*, *Salmonella give*, and *Salmonella kentucky*. Gast and Regmi showed that the isolations of *Salmonella enteritidis* in visceral parenchyma organs, such as the liver and spleen, were higher than from the cecum [25]. The results of this study showed that more birds were positive for *Salmonella pullorum* in the samples of mixed organs and cecum when compared with the sample of cloaca swabs in the challenged/untreated subgroup B1, and the same trend was also observed in the *Salmonella give*- and *Salmonella kentucky*-challenged groups B2 and B3. These results are in agreement with previous study [26]. *Salmonella* exhibited a high affinity for the intestinal tract, especially the cecum and ileocecal junction. Intestinal colonization is a pivotal first step in the process of *Salmonella*-caused disease [27]. Intestinal colonization by *Salmonella* is usually followed by invasion to the intestinal epithelium and macrophages. *Salmonella* has evolved a highly specialized membrane-bound compartment (*Salmonella*-containing vacuole, SCV), which enables *Salmonella* to avoid the phagosome/lysosome of macrophage fusion for survival and multiplication [28]. Survival of *Salmonella* within macrophages plays an important role in the dissemination of *Salmonella* to the internal organs [29]. The higher isolations of *Salmonella* in the samples of mixed organs and cecum may be attributed to these mechanisms. Therefore, mixed organs and the cecum are usually preferred for *Salmonella* isolation.

In order to fully understand the effect of SCEO against the infections of *Salmonella pullorum*, *Salmonella give*, and *Salmonella kentucky* in birds, we compared the isolations of *Salmonella* from same samples of birds in different treatment groups. The results of this study showed that the SCEO used in this study had the effect of slightly but not significantly reducing the colonization of *Salmonella pullorum* and *Salmonella give* in the samples of mixed organs, cecum, and cloaca. Interestingly, SCEO could slightly but not significantly reduce the colonization of *Salmonella kentucky* in the mixed organs, but the colonization of *Salmonella kentucky* in the cecum and cloaca was not reduced, instead slightly but not significantly increasing. The high isolation of *Salmonella kentucky* in the cecum sample may be the reason for *Salmonella kentucky* having the advantage of colonization over poultry-adapted serovars, such as *Salmonella enteritidis*, in the chicken intestines [7]. Two core *Salmonella* genes (*mgl* and *csg*), at the transcriptional level, are significantly upregulated in *Salmonella kentucky* grown in medium with cecal contents when compared to medium with glucose. These genes were differentially regulated in *Salmonella kentucky* grown in medium with cecal contents when compared to *Salmonella typhimurium* grown under similar conditions, and the differential regulation of gene *csg* was most prominent [30]. Galactose is a major nutrient that can support *Salmonella* growth, and the high-affinity galactose transporters encoded by *mglCA* and *mglB* are responsible for galactose uptake and utilization. The formation of biofilms plays a very important role in the process of persistent infection by pathogens. The curli proteins, which are adhesive fimbrial structures encoded by *csgDEFG* and *csgBAC*, are responsible for biofilm formation on biotic or abiotic surfaces, representing a major component of the extracellular matrix in biofilms [30]. Trans-cinnamaldehyde was shown to reduce the colonization ability of *Salmonella typhimurium* in the digestive tract [31]. However, SCEO used in this study had the effect of reducing the colonization of *Salmonella pullorum* and *Salmonella give* but not *Salmonella kentucky* in the cecum. This may be attributed to the differential expressions of *mgl* and *csg* genes in *Salmonella kentucky* grown in a simulated intestinal environment when compared to *Salmonella typhimurium* grown under similar conditions. *Salmonella* in the intestinal tract can be excreted from the body through feces [27]. Therefore, the higher isolation in the sample of cloaca swab in subgroup A3 when compared to subgroup B3 may be due to the fact that *Salmonella kentucky* in the intestine was excreted in feces, with SCEO promoting the excretion of *Salmonella kentucky*. These may be the reasons for the SCEO having no significant effect against the infection of *Salmonella kentucky* in birds.

*Salmonella* infection generally cause a range of symptoms, including pasty diarrhea, anorexia, emaciation, depression, and even death in very young chicks or poults, which is infrequent in mature poultry [26,32]. In this study, birds in the challenged/untreated groups exhibited weakness, poor growth, white diarrhea, anal paste, and severe liver and heart lesions. Only one bird challenged with *Salmonella pullorum* died in subgroup B1 during the experiment. These results are in agreement with a previous study [26]. This also proves that our experimental modeling was successful, and the *Salmonella* isolates could produce obvious lesions on Yellow chickens, indicating the urgency and importance of preventing and controlling *Salmonella* infection in Yellow chickens. *Salmonella pullorum* infection in chicks often leads to acute systemic diseases and is correlated with a high mortality rate [5]. The result of lower mortality may be due to the bird protective microbiota acquired from feed and environment when challenged at 7 days of age, which leads to their resistance to *Salmonella* at a certain level.

The growth retardation following *Salmonella* infection has previously been reported [26]. Decreased villus height and increased crypt depth result in fewer absorbed nutrients. The excessive consumption of nutrients by the organism to fight infection results in an increase in the amount of nutrients allocated to the immune system [33]. In this study, the results of the BW test indicated that *Salmonella* infection inhibited the normal growth of Yellow chickens. On the other hand, SCEO supplemented via drinking water could significantly improve the decrease in BW caused by the infections of *Salmonella pullorum*, *Salmonella give*, and *Salmonella kentucky* in birds, with the best effect against *Salmonella give*. This is also in agreement with the results reported in previous studies that trans-anethole and trans-cinnamaldehyde can effectively enhance the body weight gain in broilers [33]. The beneficial effects of SCEO on the BW of birds may be related to the antioxidant effects of trans-cinnamaldehyde and trans-anisole, as well as the stimulating effects on the digestive system, such as stimulating appetite, increasing the secretion of digestive enzymes and intestinal mucosa, improving intestinal morphology, and stabilizing the balance of intestinal microorganisms [17,34].

## 4. Materials and Methods

### 4.1. Materials

SCEO product, composed of trans-cinnamaldehyde (1–6 g/L) and trans-anethole (0.2–6 g/L), was kindly supplied by Guangxi GuiYuanSen Biotechnology Co., Ltd. (GuiYuan, China) One day old chicks of the Yellow chicken breed, from a PD- and PI-negative grandparent flock, were kindly supplied by Guangxi Zhu’s Agricultural and Animal Husbandry Co., Ltd. (Yulin, China) A commercial diet without any antibiotics was kindly supplied by Guangxi Zhu’s Agricultural and Animal Husbandry Co., Ltd. Mueller-Hinton (MH) Broth, MH Agar, Buffered Peptone Water (BPW), and Selenite Cystine Broth (SC) were purchased from Beijing Land Bridge Technology Co., Ltd. (Beijing, China) Xylose Lysine Desoxycholate (XLD) Agar was purchased from Guangdong Huankai Microbial Sci.&Tech. Co., Ltd. (Guangzhou, China) *Salmonella* A to F group-specific diagnostic sera and *Salmonella* serotype-specific monovalent sera were purchased from the Ningbo Tianrun Bio-pharmaceutical Co., Ltd. (Ningbo, China). Reference strains of *Salmonella pullorum*, *Salmonella give*, and *Salmonella kentucky* used for the challenge were provided by the Institute for Poultry Science and Health, Guangxi University, China.

### 4.2. Determination of Minimum Inhibitory Concentration (MIC)

Determination of the MIC of SCEO was performed using the broth microdilution method [35]. The standard inoculum was prepared in sterile NaCl solution (0.9% *w*/*v*), and it was derived from the live colonies of the selected bacteria and contained in MH agar plates with an optical density equal to 0.5 McFarland standard (corresponding to approximately 1 × 10^8^ CFU/mL). Subsequently, the inoculum was diluted at 1:100 to obtain 10^6^ CFU/mL of inoculum. Next, 100 µL of SCEO was added to the ELISA plates, and serial dilutions (1:1) were performed with MH broth to obtain different dilution multiples (2, 4, 8, 16, 32, 64, 128, 256, and 512). Then, the same volume of inoculum was added into each well. A positive control (100 µL of MH broth + 100 µL of inoculum) and a negative control (200 µL of MH broth) were established, and three replicates were set for each treatment combination. Finally, ELISA plates were cultured at 37 °C for 24 h for *Salmonella*, and the absorbance was measured at 600 nm. The MIC analyses were performed according to the difference between OD values before and after culture. The difference between OD values before and after culture was less than 0.05, indicating that the concentration was able to inhibit bacterial growth; the lowest concentration of SCEO that inhibited bacterial growth was considered the MIC.

### 4.3. Preparation of Salmonella Inoculum

The 50% infection dose (ID50) of *Salmonella* in birds was determined according to the previous preliminary test method of our laboratory [26]. The Salmonellae count per milliliter of medium was confirmed using the plate-count method. Finally, the ID50 values of *Salmonella pullorum*, *Salmonella give*, and *Salmonella kentucky* were determined as 1.8 × 10^9^ CFU/bird, 1.2 × 10^10^ CFU/bird, and 3 × 10^10^ CFU/bird, respectively.

### 4.4. Laboratory Challenge-Protection Experiment

A total of 70 1 day old chicks were divided into three groups: Group A (30 specimens) served as the treated/challenged group; group B (30 specimens) served as the untreated/challenged group; group C (10 specimens) served as the untreated/unchallenged control (see Table 2). During the entire trial of 35 days, group A was fed a commercial diet (without any antibiotics, see Table 3 [26]) and was supplemented continually with SCEO (diluted 800-fold for use according to the product instructions, 1.25 mL/L) in the drinking water. Groups B and C were fed the same commercial diet as group A but water without SCEO. At 7 days of age, birds in groups A and B were divided into three subgroups in equal number, designated as subgroups A1, A2, A3, B1, B2, and B3. Subgroups A1 and B1 were infected orally with *Salmonella pullorum* (1.8 × 10^9^ CFU/bird), subgroups A2 and B2 were infected orally with *Salmonella give* (1.2 × 10^10^ CFU/bird), subgroups A3 and B3 were infected orally with *Salmonella kentucky* (3 × 10^10^ CFU/bird), and group C orally received sterilized normal saline in the same volume.

### 4.5. Animals Management

Bird groups were separately distributed into pens of identical size in different rooms of the same building. Each pen provided 1.2 m^2^ floor space with a stocking density of 20.8 birds/m^2^ for birds. Pens was equipped with one drink dispenser and one feeder. The ambient temperature was maintained at 35 °C during the first week, and thereafter gradually reduced to 27 °C at the rate of 2 °C per week, before remaining at this temperature until the end of the experiment. Lighting was 24 h for the first week using 30 W heat lamps, followed by 15 W incandescent lamps for 22 h through the 14th day, 21 h through the 21st day, and 14 h until the end of experiment. Natural light alone was used during daylight hours after 14 days of age. All birds were given feed and water ad libitum throughout the experiment.

### 4.6. Clinical Symptoms, Pathological Changes, and Body Weight Observed and Recorded

The clinical symptoms of the experimental birds and pathological changes of the necropsied birds were observed and recorded daily. At the end of the experiment at 35 days of age, the live BW of all the birds in each group was recorded.

### 4.7. Recovery of the Challenged Salmonella

At the end of the experiment at 35 days of age, all surviving birds were euthanized, and the organs were excised aseptically. Samples of cloaca swabs, cecum, and mixed visceral parenchyma organs including the heart, liver, and spleen were taken for the isolation of the challenged *Salmonella* according to the routine of China National Food Safety Standard Methods for Food Microbiological Examination-*Salmonella* (GB/T4789.4-2016). According to this examination procedure, 1 g of homogenized sample was first pre-enriched in 10 mL of BPW. The mixture was incubated at 37 °C for 24 h. Then, 1 mL of sample, previously incubated in BPW, was transferred into 10 mL of SC for selective enrichment and incubated at 37 °C for 24 h. After that, one loopful of broth was streaked onto XLD agar and incubated at 37 °C for 24 h. Then, presumptive bacterial colonies on XLD agar were detected with A to F *Salmonella* diagnostic sera using the slide agglutination test. Next, positive bacterial strains were identified by PCR using *Salmonella*-specific identification primers, and the PCR products were analyzed using 1.5% agarose gel electrophoresis. Lastly, *Salmonella* isolates were further classified by *Salmonella* serotype-specific monovalent sera, according to the motility test and a biochemical test.

### 4.8. Statistical Analysis

Microsoft Excel and the Statistical Program for Social Sciences (SPSS) 22.0 statistical software were used to process the experimental data. Factor analysis of variance and the chi-square test were used to test the difference between groups. A *p*-value < 0.05 was used to indicate statistical significance.

## 5. Conclusions

In summary, SCEO is effective against *Salmonella pullorum*, *Salmonella give*, and *Salmonella kentucky* in vitro, and SCEO supplemented via drinking water could significantly reduce the infections of *Salmonella pullorum* and *Salmonella give* after the challenge in Yellow chickens, while slightly but not significantly reducing the infection of *Salmonella kentucky*; SCEO could also significantly improve the BW decline caused by these *Salmonella* infections. SCEO had the best effect against the infection of *Salmonella give* in Yellow chickens, followed by *Salmonella pullorum* and *Salmonella kentucky*. Therefore, SCEO supplemented in the drinking water could be used as a preventive strategy against the infection of *Salmonella*.

## Figures and Tables

**Figure 1 antibiotics-11-01579-f001:**
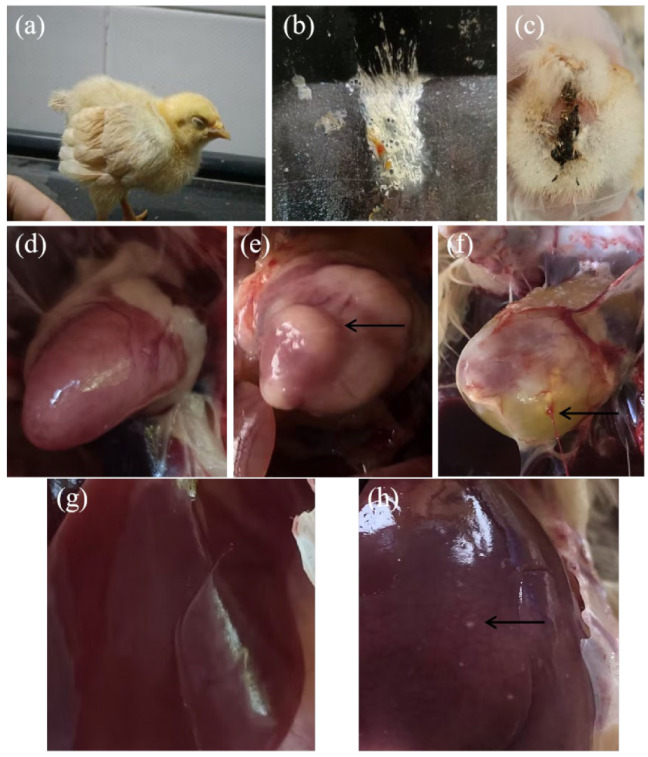
The clinical signs and gross lesions of the challenged birds observed. (**a**) Depression and poor growth; (**b**) diarrhea; (**c**) wet paste; (**d**) heart with no lesion; (**e**) heart with granuloma; (**f**) soft heart with pericardial effusion; (**g**) liver with no lesion; (**h**) pinpoint white foci of necrosis on liver surface.

**Figure 2 antibiotics-11-01579-f002:**
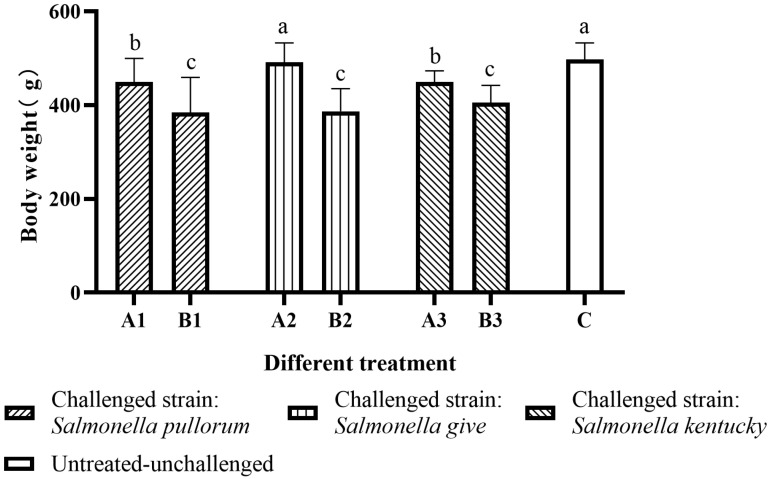
Body weight of birds in different groups. The means with different letters denote a significant difference (*p* < 0.05).

**Figure 3 antibiotics-11-01579-f003:**
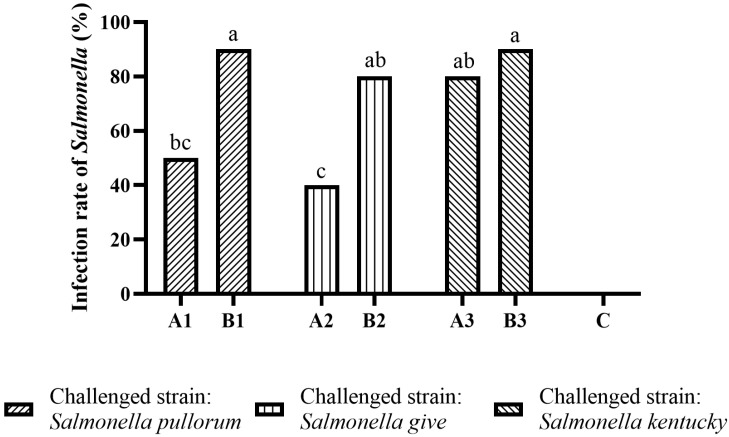
The infection rates of *Salmonella* in the birds in the *Salmonella pullorum*, *Salmonella give*, and *Salmonella kentucky*-challenged groups. The means with different letters denote a significant difference (*p* < 0.05).

**Figure 4 antibiotics-11-01579-f004:**
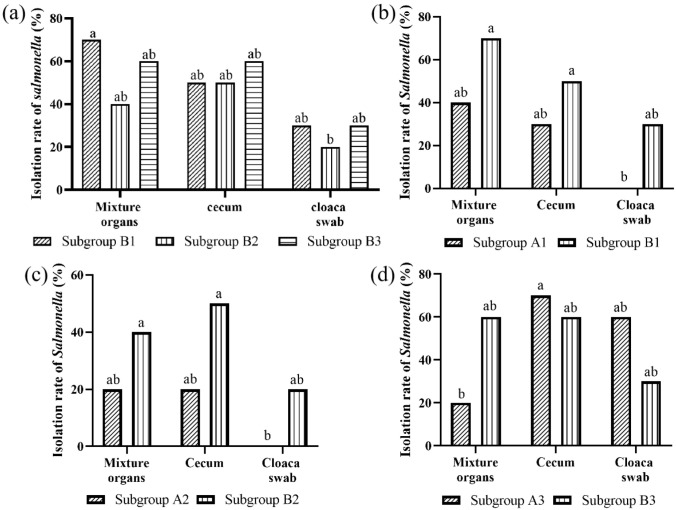
(**a**) The isolations of *Salmonella* from different samples of birds in challenged/untreated groups. (**b**) The isolations of *Salmonella pullorum* from different samples of birds in the *Salmonella pullorum*-challenged groups. (**c**) The isolations of *Salmonella give* from different samples of birds in the *Salmonella give*-challenged groups; (**d**) The isolations of *Salmonella kentucky* from different samples of birds in the *Salmonella kentucky*-challenged groups. The means with different letters denote a significant difference (*p* < 0.05).

**Table 1 antibiotics-11-01579-t001:** MIC of SCEO.

*Salmonella* Strains	Dilution (SCEO ^1^)								
	1:2	1:4	1:8	1:16	1:32	1:64	1:128	1:256	1:512
*Salmonella pullorum*	−	−	−	+	+	+	+	+	+
*Salmonella give*	−	+	+	+	+	+	+	+	+
*Salmonella kentucky*	−	+	+	+	+	+	+	+	+

“−”—dilution was effective against analyzed bacteria, “+”—dilution was not effective against analyzed bacteria. ^1^ SCEO: trans-cinnamaldehyde (1–6 g/L) and trans-anethole (0.2–6 g/L).

**Table 2 antibiotics-11-01579-t002:** Arrangement of laboratory challenge/protection experiment.

Groups		Treatments	Challenge Strains	No. of Birds
A	A1	SCEO (1.25 mL/L supplemented in the water)	*Salmonella pullorum*	10
	A2		*Salmonella give*	10
	A3		*Salmonella kentucky*	10
B	B1	No SCEO	*Salmonella pullorum*	10
	B2		*Salmonella give*	10
	B3		*Salmonella kentucky*	10
C	Untreated/unchallenged	No SCEO	-	10

**Table 3 antibiotics-11-01579-t003:** Composition of basal diet used in the experiment.

Items (g/kg)	Starter ^3^	Grower A ^4^	Grower B ^5^
Corn	596	688	750
Soybean meal (78.0 g/kg)	342	266	214
Soybean oil	22	9	0
Limestone (370 g/kg)	12.7	13.4	13.2
Calcium hydrogen phosphate	18	15	15
Sodium chloride	2.5	2.5	2.5
Methionine (998 g/kg)	2.5	2	1.6
Lysine HCL (780 g/kg)	2	1.8	1.4
Vitamin premix ^1^	0.3	0.3	0.3
Mineral premix ^2^	1	1	1
Choline chloride (500 g/kg)	1	1	1
Total	1000	1000	1000
Calculated nutrient content			
Metabolizable energy (kcal/g)	2950	2950	2920
Crude protein (g/kg)	198	171	152
Calcium (g/kg)	0.95	0.95	0.95
Available phosphorus (g/kg)	4.5	4.5	4
Lysine (g/kg)	12.1	10	8.6
Methionine (g/kg)	5.4	4.6	4

^1^ Vitamin premix supplied the following per kg of diet: 10,500 IU of vitamin A, 3000 IU of vitamin D3, 25 IU of vitamin E, 0.03 mg of vitamin B12, 0.15 mg of biotin, 4 mg of menadione (K3), 4 mg of thiamine, 6 mg of riboflavin, 18 mg of D-pantothenic acid, 6 mg of vitamin B6, 50 mg of niacin, and 1.5 mg of folic acid. ^2^ Mineral premix supplied the following per kg of diet: manganese, 100 mg; zinc, 95 mg; iron, 95 mg; copper, 12 mg; iodine, 0.8 mg; sodium selenite 0.2 mg. ^3^ The starter diet was fed from approximately 0 to 42 days of age. ^4^ The grower A diet was fed from approximately 43 to 84 days of age. ^5^ The grower B diet was fed from approximately 85 to 120 days of age.

## Data Availability

Not applicable.

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
