# Peer review of "The Use of Star Anise-Cinnamon Essential Oil as an Alternative Antibiotic in Prevention of Salmonella Infections in Yellow Chickens"

_antibiotics, 2022, doi:10.3390/antibiotics11111579_

Round 1

Reviewer 1 Report

it is good paper and I have comments

1- can you use references from middle east?

2- Do you plan or arrange to use another oil of herbal or natural oil instade of Star anise-cinnamon essential oil?

I thing its good idea to compare between them

3- Do you have any idea if SCEO can treat pathogenic bacteria (infected human) ?

4- Is cinnamon oil pure active or the trans-cinnamaldehyde is the active part of this oil? to treat infection generally

Author Response

We are very grateful for your comments regarding the manuscript entitled “The Star anise-cinnamon essential oil can be used as an alternative antibiotic to prevent the Salmonella infection in Yellow-chickens”. All your comments and suggestions are very important to us, both for composing the manuscript and our further research. We have studied comments carefully and have made corrections which we hope meet with approval.

Based on your advice, we amended the relevant section in the manuscript. All your questions are answered below.

1-can you use references from middle east?

Response:

The references from middle east have been cited in manuscript.

Lines 460-462

Lines 468-470

Lines 495-496

Lines 497-499

2-Do you plan or arrange to use another oil of herbal or natural oil instead of Star anise-cinnamon essential oil? I thing its good idea to compare between them.

Response:

Your idea is very good, but we have not plan or arrange to study another oil of herbal or natural oil recently. Your ideas provide new thought for our future research, we are going to continue to explore difference between other oil of herbal or natural oil and Star anise-cinnamon essential oil in future follow-up papers.

3-Do you have any idea if SCEO can treat pathogenic bacteria (infected human) ?

Response:

Research shows that Trans-cinnamaldehyde has antibacterial effect against food-borne pathogenic bacteria, such as Salmonella typhimurium, Salmonella enteritidis and Listeria monocytogenes and so on. In vitro antibacterial activity of the newly added SCEO against Salmonella in our manuscript shows that SCEO has also antibacterial effect against food-borne pathogenic bacteria (Salmonella give and Salmonella kentucky). Therefore, we believe that SCEO could treat food-borne pathogenic bacteria.

4-Is cinnamon oil pure active or the trans-cinnamaldehyde is the active part of this oil? to treat infection generally.

Response:

The trans-cinnamaldehyde is the active part of cinnamon oil to treat general infection. But the presence of some minor compounds present with the major compound trans-cinnamaldehyde could enhance the effect of cinnamon oil.

Reviewer 2 Report

The paper is an in vivo study on the anti Salmonella effect of a mixture of Eos from star anise and cinnamon. These eos are well known compounds effective against such microorganisms, but the study is novel for its formulation (preventive in vivo action). Some information should be added about the selected oils

Please, delete the abbreviation in brackets i.e. S. pullorum) this is not correct

lines 70 and following - oils are

please provide the complete composition of Eos employed and cite the following papers

doi: 10.1007/s00284-022-02938-x.

doi: 10.3390/molecules26185598.

doi: 10.1038/s41598-022-13511-8

doi: 10.3390/molecules24050900.

doi: 10.3390/foods11152234.

Author Response

We are very grateful for your comments regarding the manuscript entitled “The Star anise-cinnamon essential oil can be used as an alternative antibiotic to prevent the Salmonella infection in Yellow-chickens”. All your comments and suggestions are very important to us, both for composing the manuscript and our further research. We have studied comments carefully and have made corrections which we hope meet with approval.

Based on your advice, we amended the relevant section in the manuscript. All your questions are answered below.

The paper is an in vivo study on the anti Salmonella effect of a mixture of Eos from star anise and cinnamon. These eos are well known compounds effective against such microorganisms, but the study is novel for its formulation (preventive in vivo action).

1-Some information should be added about the selected oils.

Response:

Some informations have been added about the selected oils.

Lines 298-299

2-Please, delete the abbreviation in brackets i.e. S. pullorum) this is not correct. lines 70 and following - oils are.

Response: 

We apologize for these grammatical problem and have corrected them based on your suggestions.

Line 17

Line 35

Line 37

3-please provide the complete composition of Eos employed and cite the following papers.

doi: 10.1007/s00284-022-02938-x.

doi: 10.3390/molecules26185598.

doi: 10.1038/s41598-022-13511-8

doi: 10.3390/molecules24050900.

doi: 10.3390/foods11152234.

Response:

Based on your advice, we have cited the recommended papers in manuscript. The research and development of EOs takes into account the copyright issue and is not convenient to disclose other specific information. Therefore, we seek for the your tolerance and understanding. Many thanks for your kind help !

Lines 469-470

Lines 506-508

Lines 467-468

Lines 477-478

Lines 479-480

Reviewer 3 Report

The comments and suggestions for authors are provided in attached file.

Author Response

We are very grateful for your comments regarding the manuscript entitled “The Star anise-cinnamon essential oil can be used as an alternative antibiotic to prevent the Salmonella infection in Yellow-chickens”. All your comments and suggestions are very important to us, both for composing the manuscript and our further research. We have studied comments carefully and have made corrections which we hope meet with approval.

Based on your advice, we amended the relevant section in the manuscript. All your questions are answered below.

The comments and suggestions for authors are provided in attached file.

Response:

Thank you for your comments and suggestions. We apologize for these grammatical problems and have corrected it based on your suggestions. We hope manuscript is now clearer.

Reviewer 4 Report

The submitted article cannot be published in its current form. It requires major changes.

1. The authors need to improve and complete the study methods.

2. Many important parameters are missing from the publication. The authors need to perform a biochemical analysis of the infected organs and compare the results to the control group.

3. Please provide the exact morphological changes of the infected organs compared to the control groups.

4. The results are poorly described. Graph 2 is illegible.

5. The in vitro antibacterial activity of SCEO against the Salmonella strains used could be added to the publication (determined MIC- minimum inhibitory concentration for bacterial growth).

6. Please correct editorial errors in the text, e.g. no line spacing; Table 1 has the abbreviation SECO, should be SCEO.

Author Response

We are very grateful for your comments regarding the manuscript entitled “The Star anise-cinnamon essential oil can be used as an alternative antibiotic to prevent the Salmonella infection in Yellow-chickens”. All your comments and suggestions are very important to us, both for composing the manuscript and our further research. We have studied comments carefully and have made corrections which we hope meet with approval.

Based on your advice, we amended the relevant section in the manuscript. All your questions are answered below.

The submitted article cannot be published in its current form. It requires major changes.

1-The authors need to improve and complete the study methods.

Response: 

We apologize for the deficient study methods and have improved and completed it.

Lines 345-358

Lines 390-400

2-Many important parameters are missing from the publication. The authors need to perform a biochemical analysis of the infected organs and compare the results to the control group.

Response:

We appreciate the reviewer’s insightful suggestion and agree that it would be useful to support our conclusion. However, such an analysis is beyond the scope of our paper, and is not the important aspect of our paper. For this reason, we chose not to make this change. Therefore, we seek for the reviewer’s tolerance and understanding. Many thanks for your kind help !

3-Please provide the exact morphological changes of the infected organs compared to the control groups.

Response:

We apologize for that the lack of morphological figure of infected organs in the control group. We have added the morphological figure of infected organs in the control group and hope that it is now clear that exact morphological changes of infected organs between challenged group and control group.

Line 113

4-The results are poorly described. Graph 2 is illegible.

Response: 

We apologize for that the results are poorly described and Graph 2 is illegible, We have revised the results and modified the Graph 2 to address your concerns and hope that them are now clearer. 

Lines 131-138

Lines 142-156

Line 139

Line 157

5-The in vitro antibacterial activity of SCEO against the Salmonella strains used could be added to the publication (determined MIC- minimum inhibitory concentration for bacterial growth).

Response: 

Your suggestion is very reasonable. We have added the antibacterial activity of SCEO against the Salmonella strains in vitro to the manuscript. 

Lines 98-104

6-Please correct editorial errors in the text, e.g. no line spacing; Table 1 has the abbreviation SECO, should be SCEO.

Response:

We apologize for these grammatical problems and have corrected them based on your suggestions.

Line 359

Round 2

Reviewer 2 Report

The manuscript is improved after the revision and can be accepted

Reviewer 4 Report

Once changes have been made, the manuscript can be published.